# Efficacy of a Multicomponent Intervention for Fibromyalgia Based on Pain Neuroscience Education, Exercise Therapy, Psychological Support, and Nature Exposure (NAT-FM): Study Protocol of a Randomized Controlled Trial

**DOI:** 10.3390/ijerph17020634

**Published:** 2020-01-19

**Authors:** Mayte Serrat, Juan P. Sanabria-Mazo, Elna García-Troiteiro, Anna Fontcuberta, Corel Mateo-Canedo, Míriam Almirall, Albert Feliu-Soler, Jorge Luis Méndez-Ulrich, Antoni Sanz, Juan V. Luciano

**Affiliations:** 1Unitat d’Expertesa en Síndromes de Sensibilització Central, Hospital de la Vall d’Hebron, 08035 Barcelona, Spain; mserrat@vhebron.net (M.S.); malmirall@vhebron.net (M.A.); 2Research Group on Stress and Health, Faculty of Psychology, Universitat Autònoma de Barcelona, 08193 Cerdanyola del Vallès, Spain; juanpablo.sanabria@e-campus.uab.cat (J.P.S.-M.); elna.garcia@e-campus.uab.cat (E.G.-T.); anna.fontcu@gmail.com (A.F.); corel.mateo@e-campus.uab.cat (C.M.-C.); a.feliu@pssjd.org (A.F.-S.); 3Escoles Universitàries Gimbernat, Universitat Autònoma de Barcelona, Sant Cugat del Vallès, 08174 Barcelona, Spain; 4Institut de Recerca Sant Joan de Déu, 08950 Esplugues de Llobregat, Spain; 5Teaching, Research, & Innovation Unit - Parc Sanitari Sant Joan de Déu, 08830 St. Boi de Llobregat, Spain

**Keywords:** fibromyalgia, multicomponent treatment, pain neuroscience education, exercise therapy, cognitive behavioural therapy, nature exposure, randomized controlled trial, ecological momentary assessment, study protocol

## Abstract

The study protocol of a prospective and randomized controlled trial for the assessment of the efficacy of nature activity therapy for people with Fibromyalgia (NAT-FM) is described. The primary outcome is the mean change from baseline in the Revised Fibromyalgia Impact Questionnaire (FIQR) score at post-treatment (12 weeks) and at 9 months of follow-up, and secondary outcomes are changes in the positive affect, negative affect, pain, fatigue, self-efficacy, catastrophising, and emotional regulation. A total of 160 patients with fibromyalgia will be divided into two arms: treatment-as-usual (TAU) and NAT-FM+TAU. Pre, during, post, +6, and +9 months assessments will be carried out, as well as an ecological momentary assessment (EMA) of intrasession and intersessions. Results will be subjected to a mixed group (NAT-FM+TAU vs. TAU) × phase (pre, post, +6 months, +9 months) general linear model. EMA intrasession measurements will be subjected to a 2 (pre vs. post) × 5 (type of activity) mixed-effects ANOVA. EMA between-session measurements obtained from both arms of the study will be analysed on both a time-domain and frequency-domain basis. Effect sizes and number needed to treat (NNT) will be computed. A mediation/moderation analysis will be conducted.

## 1. Introduction

According to the American College of Rheumatology (ACR), Fibromyalgia (FM) is a chronic disease of unknown etiology, characterised by the presence of generalized musculoskeletal pain and other symptoms, such as fatigue, waking unrefreshed, and cognitive problems [1]. The etiopathogenesis of FM involves several mechanisms, the most notable of these being the sensitization of the central nervous system (CNS). Abnormalities in the ascending and descending pathways associated with pain processing have been observed in FM [2]. The chronic nature of pain, wide range of symptoms, and comorbidity with mental disorders (mainly anxiety and depression) strongly impact on the well-being and quality of life of patients with FM [1,3,4].

Currently, the prevalence of FM among the general population is estimated to be around 2% worldwide and 2.6% in Europe [5]. In Spain, a prevalence of 2.4% is reported in the general population over 20 years of age, corresponding to 4.2% for women and 0.2% for men [6]. The direct costs (medical care, prescription of medicines, etc.) and indirect costs (absenteeism from work, loss of work, etc.) of FM are close to 11 billion € per year [7]. Due to the variety of complex factors involved in FM, recent years have seen the development and testing of various pharmacological [8] and non-pharmacological [9,10] treatments for management of the condition. The side effects associated with pharmacological treatments, along with their suboptimal clinical effects, have recently prompted the European League against Rheumatism (EULAR) to recommend that these treatments should only be used to control pain and sleep disturbances caused by FM [11].

Multicomponent treatments integrating exercise therapy (ET), cognitive behavioural therapy (CBT), and pharmacological treatment have generally been one of the most beneficial alternatives for improving quality of life, increasing functional capacity, and decreasing chronic pain in FM patients [12]. Randomized controlled trials (RCTs), testing for the simultaneous combination of ET and CBT, have been shown to be the most effective non-pharmacological treatments for FM [13]. Treatments based exclusively on ET have contributed to the reduction of symptoms of pain, fatigue, and depression, as well as to the improvement of mental health, psychological well-being, and physical function [14,15]. CBT-based treatments have favoured the acceptance of FM, strengthening of self-efficacy, development of coping strategies, and the reduction in depressive moods [16]. Pain neuroscience education (PNE) has been shown to be more effective than biomedical education [17] for FM and it is proving to be a key strategic piece in both the physical and cognitive approach of FM patients [18,19,20].

The strengths of such treatments, coupled with the identified benefits of exposure to natural contexts for people’s mental health, justify the need to explore new interventions for FM patients conducted in nature. Considering the moderate benefits that have been reported for the four therapeutic components mentioned above, it seems necessary to test empirically, for the first time, the synergistic effects of its combination. There is some evidence that nature can exert a synergistic effect when added to other therapeutic ingredients [21,22,23,24,25,26]. In this context, the Nature, Activity Therapy for people with Fibromyalgia (NAT-FM) project has emerged as a new generation of therapeutic treatments for intervention of health problems, which integrates CBT, PNE [27], ET, and nature exposure. Although activity-based interventions in nature have demonstrated positive effects for treating emotional, cognitive, and behavioural functioning in both the general population [28,29,30] and clinical population [31,32], as far as we know there is only one clinical trial that provides empirical evidence of their efficacy in FM patients [33]. 

This paper describes the design characteristics of the NAT-FM treatment in comparison with a treatment-as-usual (TAU). The objectives of the RCT are the following: (a) to analyse the efficacy of the NAT-FM program as an add-on to TAU in improving the functional status (primary outcome) of FM patients; (b) to compare the efficacy of NAT-FM treatment with respect to TAU in improving affectivity, emotional regulation, perceived competence, self-esteem, self-efficacy, anxiety, depression, and other secondary outcomes; and (c) to identify factors that may act as mediators or moderators of the efficacy of NAT-FM treatment (age, years of evolution, psychological inflexibility, etc.). 

## 2. Materials and Methods 

### 2.1. Research Design

The RCT protocol was developed following the Standard Protocol Items: Recommendations for Interventional Trials (SPIRIT) [34] and has been registered with ClinicalTrials.gov (Trial registration: NCT04190771). This study will employ a 9-month, randomized controlled trial (RCT) design, with two treatment arms. For the reporting of the RCT, we will follow the guidelines of the Consolidated Standards of Reporting Trials (CONSORT) [35]. The two arms of the treatment will be: (a) TAU (control group) and (b) TAU+NAT-FM (intervention group). Patients in both arms received TAU, since NAT-FM will, in principle, be a complementary treatment to that which is usually provided by the Spanish National Health System [36].

### 2.2. Participants

A total of 160 patients with FM will be recruited from the Central Sensitivity Syndromes Unit (CSSU) at the Vall d’Hebron University Hospital (Barcelona, Spain). The sample size was established using IBM SPSS SamplePower 3.0 (IBM, Chicago, IL, USA), considering the results of a meta-analysis of therapies in nature [21], which reported an average effect size of *g* = 0.50, regarding clinical variables. With the anticipation of a 20% dropout and setting alpha = 0.05 and power 1-b = 0.80, the estimated sample size is *n* = 80 participants per condition. The sample size in each of the arms and the treatment follow-up period were estimated in such a way that they will be sufficiently representative to capture important results.

Patients visited consecutively by the physical therapist of the CSSU from November to December 2020 who met the selection criteria will be recruited. The inclusion criteria will be: (a) adults ≥18 years old, (b) to meet the 2010–2011 ACR diagnostic criteria for FM [1,37,38], and (c) to be able to understand and agree to participate in the study. The exclusion criteria will be: (a) participating in concurrent or past RCTs (previous year) and (b) exhibiting comorbidity with severe mental disorders (e.g., psychosis) or neurodegenerative diseases (e.g., Alzheimer’s) that would limit the ability of the patient to participate in the RCT.

### 2.3. Procedure

The main researcher (M.S.) will, through an initial interview, provide an overview of the study to FM patients interested in participating who met inclusion and exclusion criteria. Prior to being randomly allocated to treatments (TAU or TAU+NAT-FM), informed consent forms will be collected, which will include a detailed description of the characteristics of the interventions. Patients will also be informed that their participation will be voluntary and that they will be able to withdraw at any time, with the guarantee that they will continue to receive their usual treatment. This research will be conducted in accordance with the ethical standards set forth in the 1964 Declaration of Helsinki and was approved by the hospital’s Ethics Committee (PR(AG)120/2018). Patient data will be treated confidentially, ensuring that only the research team can access this information after recoding the name and personal identity number. Only the principal investigator will have access to the patient code key, which, according to current data protection legislation in Spain, will be stored separately in a safe place. The participants will be assigned to the intervention group (TAU+NAT-FM) or control group (TAU), employing a SPSS v25 randomization list.

The pharmacological treatment will not be modified during the RCT. Only rescue paracetamol (maximum 1 gram every 8 hours) will be allowed if there is worsening of pain. Patients will be evaluated before (pre), after the 6th session of treatment (during), and after (post) treatment, as well as 6 and 9 months after baseline assessment (follow-up). Measurements shall be made according to the time established for their administration: (a) Classical structural assessment (CSA): pre, during, post, and follow-up; and (b) ecological momentary assessment (EMA): intrasession (session log) and intersessions (day log between sessions). The flow chart of the RCT is presented in Figure 1. 

#### Forecast Execution Dates

Recruitment of patients: November 2019Finalisation of patient monitoring period: July 2020Publication of results: June 2021

### 2.4. Treatments

#### 2.4.1. Intervention Group (TAU + NAT-FM)

NAT treatments are a new generation of therapeutic programs for health intervention, in which CBT, PNE [27], ET, and nature exposure are integrated. NAT-FM is conceived as an add-on therapy and, in particular, has the primary therapeutic objective of helping patients to improve their functional status. The secondary objectives are to contribute to improving their affectivity, emotional regulation, and self-efficacy, as well as to reduce their perception of pain, fatigue, and catastrophising. Other variables that are usually improved by these treatments are perceived competence, self-esteem, stress, sleep quality, anxiety, depression, psychological inflexibility, kinesiophobia, physical function, and functionality. Table 1 and Table 2 show an outline of the NAT-FM.

The NAT-FM design process was guided by the procedures established in the protocol and the proof of concept study. The therapeutic objectives were selected by taking into account the results of a systematic review of the psychological characteristics of people with FM (affective, cognitive, metacognitive, and personality profiles) [39] and the validation of a panel of experts made up of researchers and psychologists (M.S., J.P.S.-M., A.F.-S., J.L.M.-U., J.V.L, and A.S.), who evaluated the existing scientific evidence and the clinical and investigative relevance of each objective. The panel concluded that the processes to be included in the treatment are self-efficacy, negative affect, positive affect, emotional regulation, and catastrophising. The activities indicated in Table 1 were selected considering the results of an empirical study on the therapeutic potential of 10 activities in nature [40]. The number of activities practiced in each session was adapted according to the established targets, taking into account the progression of the patients and the evolution of the treatment. The activities addressing each target were selected considering the therapeutic potential identified by the expert panel.

The sectors (natural spaces) in which the outdoor sessions will be conducted have been validated by experts to guarantee their suitability for the treatment activities [40]. The geographical areas are Sant Genís Forest (coordinates N 41° 25.930 E 2° 8.237) and Les Escletxes del Papiol (coordinates N 41° 26.319 E 2° 1.101). The researchers conducted a proof of concept study (J.P.S.-M., E.G.-T., C.M.-C., and A.S) in these sectors to test the manualisation of procedures and to evaluate the suitability of the NAT-FM structure based on the testing of a series of representative sessions [41]. 

The protocol of the NAT-FM is the result of the consensus of the proposed instruments, of the moments of assessments corresponding to each phase of the treatment, and of the administration platforms used. The decision to combine different assessments (CSA+EMA) is motivated by the need for a strategy that could obtain more precise information about the dynamics of the variables to be evaluated and, in particular, could record the affective and cognitive impact of each activity, as well as its transfer to everyday life. The frequency of administration of the EMA has been established from analysing the results of a systematic review on the characteristics of its use in studies with patients with chronic pain [42]. 

The NAT-FM treatment will be carried out in a group-based format in groups made up of a maximum of 20 patients per session with a total of 12 weekly 2 h sessions (1 session per week). The sessions will be directed by the M.S., who is a physiotherapist of the CSSU of Vall d’Hebron University Hospital and also a psychologist and sports technician with the required legal qualifications for conducting this activity.

#### 2.4.2. Control Group (TAU)

The TAU provided to patients in the control group of this study is based primarily on the prescription of drugs adjusted to the symptomatic profile of each patient, with complementary advice on aerobic exercise and pain neuroscience education adapted to the physical capacities of the patients. Patients of the control group will be placed on a waiting list so that at the end of the RCT they can benefit from the NAT-FM treatment. 

### 2.5. Study Measures

The participants of the intervention group (TAU+NAT-FM) and the control group (TAU) will complete the instruments described below. Assessment will be organized considering the timings (CSA+EMA) that were established from the results of the proof of concept study [41]. Table 3 shows the measurement scheme and timings for the NAT-FM treatment protocol.

#### 2.5.1. Classical Structural Assessment (CSA)

The CSA will measure general information, clinical characteristics and screening, primary outcomes, secondary outcomes, and additional secondary outcomes. These instruments will be applied to patients in both the control group (TAU) and intervention group (TAU+NAT-FM).

##### General Information Measures

The Socio-demographic and clinical questionnaire will be used to obtain general and clinical patient data (age, educational level, socioeconomic status, marital status, ethnic group, personal medical history, years of FM diagnosis, comorbid medical conditions, etc.).

##### Measures of Clinical Features and Screening

The Structured Clinical Interview for DSM Axis I Disorders (SCID-I) [43] will be used for the diagnosis of mood disorders. It is based on the research version of SCID-I and the DSM-IV criteria. 

##### Primary Outcome Measure 

The Revised Fibromyalgia Impact Questionnaire (FIQR) comprises three dimensions: physical dysfunction (scores from 0 to 30), overall impact (scores from 0 to 20), and intensity of the symptoms (scores from 0 to 50), which are used to measure the impact generated by FM during the last week. It consists of 21 items, which are answered on a numerical rating scale of 11 points (from 0 to 10). Total scores can range from 0 to 100, with higher scores reflecting greater deterioration. The Spanish version has an adequate internal consistency (*α* = 0.93) and acceptable test-retest reliability (*r* = 0.84) [44].

##### Secondary Outcome Measures

The Positive and Negative Affect Schedule (PANAS) is used to evaluate positive and negative affects. It consists of two dimensions (positive affect and negative affect) of 10 items, each answered on a 5-point Likert scale. Total scores of each scale range from 10 to 50, where higher scores indicate a greater presence of the specific affectivity. The Spanish version has an adequate internal consistency for the positive affect (*α* = 0.92) and for the negative affect (*α* = 0.88) [45].

The Cognitive Emotion Regulation Questionnaire (CERQ) is used to assess individual differences in the cognitive regulation of emotions. The instrument measures nine 2-item dimensions (self-blame, blaming others, acceptance, refocusing on planning, positive refocusing, rumination, positive reappraisal, putting into perspective, and catastrophising). This study will use the short 18-item version. Responses are given on a 5-point Likert scale ranging from 1 (almost never) to 5 (almost always). Total scores for each dimension range from 2 to 10, with the highest scores indicating the specific cognitive strategy most used. The Spanish version has an adequate internal consistency (*α* = 0.77 to 0.93) and acceptable test-retest reliability (*r* = 0.60 to 0.85) for the subscales [46].

The Personal Perceived Competence Scale (PPCS) is used to measure perceived competence. It consists of eight items that are answered on a 6-point Likert scale. Total scores of each scale rangefrom 8 to 48, with higher scores indicating greater perceived competence. The Spanish version has an adequate internal consistency (*α* = 0.83) [47].

The Rosenberg Self-Esteem Scale (RSES) is used to measure self-esteem. It consists of 10 items that are answered on a 4-point Likert scale. Total scores of each scale range from 10 to 40, where higher scores indicate higher self-esteem. The Spanish version has an adequate internal consistency (*α* = 0.87) and acceptable test-retest reliability (*r* = 0.72 to 0.74) [48].

The Pain Catastrophizing Scale (PCS) is used to evaluate catastrophic thoughts associated with pain. It consists of three dimensions (rumination, magnification, and helplessness) of 13 items in total, which are answered on a 5-point Likert scale. Total scores on each scale range from 0 to 52, with higher scores indicating more catastrophising. The Spanish version has an adequate internal consistency (*α* = 0.79) and acceptable test-retest reliability (*r* = 0.84) [49].

The Perceived Stress Scale (PSS) is used to evaluate the stress perceived by people during the previous month. This study will use a 4-item version with a 5-point Likert type response format. Total scores range from 0 to 16, with higher scores indicating greater perceived stress. The Spanish version has an acceptable internal consistency (*α* = 0.77) [50].

##### Additional Secondary Outcomes Measures

The Hospital Anxiety and Depression Scale (HADS) is used to quantify the severity of anxiety and depression symptoms. It consists of two dimensions (anxiety and depression) of seven items, each responding on a 4-point Likert scale. Total scores of each scale (HADS-A and HADS-D) range from 0 to 21, where higher scores indicate greater severity of symptoms. The Spanish version has an adequate internal consistency for anxiety (*α* = 0.83) and for depression (*α* = 0.87) [51,52].

The Psychological Inflexibility in Pain Scale (PIPS) is used to assess psychological inflexibility in patients with pain. It consists of two dimensions (avoidance and cognitive fusion with pain) of 12 items, which are answered on a 7-point Likert scale. Total scores of each scale range from 12 to 84, with higher scores indicating greater psychological inflexibility. The Spanish version has an adequate internal consistency (*α* = 0.90) and test-retest reliability (*r* = 0.97) [53].

The Tampa Scale for Kinesiophobia (TSK-11) is used to assess fear of pain and movement. It consists of 11 items, which are answered on a 4-point Likert scale. Total scores of each scale range from 11 to 44, where higher scores indicate a greater fear of pain and movement. The Spanish version has an adequate internal consistency (*α* = 0.79) [54].

The physical function of the 36-Item Short Form Survey (SF-36) will be used to assess the physical function typically affected in patients with chronic pain. This dimension comprises a total of 10 items, which are answered on a Likert scale of 3 points. Total scores on each scale range from 0 to 100, with higher scores indicating better physical function. The dimensions have shown optimal internal consistency (*α* = 0.94) [55].

The UKK Walk Test (UKK) [56] is used to assess people’s physical fitness, endurance, and cardio-respiratory capacity (maximum VO^2^). The test is performed in a straight line with no difference in height between the two extremes. In this study, the distance from the beginning to the end of the route will be adapted from 2 km (original version) to 1 km, given the specific characteristics of the population of this study. In addition, we will use Polar Advantage heart rate monitors with their respective software to mark the times and obtain the average heart rate of the patients during the route. 

The Adverse effects Assessment Checklist [57] is an ad hoc measure to check for potential adverse events (e.g., headaches, dizziness, physical injuries) across the interventions and follow-up.

The Patient Global Impression of Change (PGIC) [58] and Pain Specific Impression of Change (PSIC) [57] are self-reported measures frequently used as indicators of meaningful change in treatments for chronic pain. Responses are given on a 7-point Likert scale ranging from 1 (much better) to 5 (much worse). The PGIC is one item that refers to the perception of global improvement, whereas the PSIC asks about the impression of change in more specific domains (physical and social functioning, work-related activities, mood, and pain). These scales will be completed by the participants assigned to the treatment (NAT-FM+TAU) group.

#### 2.5.2. Ecological Momentary Assessment (EMA)

EMA is used to assess: (a) the specific short-term impact of each activity and (b) the transfer of the treatment effects to everyday life. Measurements during the sessions (intrasession assessment) will be taken before and after each activity in nature using an online form. Measurements of the daily records between sessions (intersession assessment) will be carried out through an app, in which patients will have to respond six times a day (twice in the morning, twice in the afternoon, and twice at night). Intersession instruments will be applied to patients in both the control group (TAU) and intervention group (TAU+NAT-FM). However, intrasession instruments will only be applied to patients in the intervention group (TAU+NAT-FM).

##### Other Secondary Outcome Measurements

The Self-Assessment Manikin (SAM) [59] is used to assess the affective state of people. It consists of three blocks of diagrammed pictograms in a continuous line, representing the following three dimensions of the affective response: valence, arousal, and dominance. Each scale has a 9-point Likert response format. The total scores on each scale range from 1 to 9. Higher scores on valence indicate greater positive mood; higher scores on arousal indicate greater activation and alertness; whilst for dominance a higher score indicates greater perception of control and personal confidence. These three dimensions of the affective response were evaluated both in intrasession assessment and in intersession assessment.

Single-item questionnaires: For the assessments of the variables fatigue, pain, and sleep quality, the three items of the Visual Analog Scale (VAS) of the FIQR were selected. For stress and self-efficacy, ad hoc questions were designed with a single item, rated on a scale from 0 to 10, in a VAS format. Higher scores indicate greater perceived fatigue, pain, sleep quality, stress, and self-efficacy. The variables fatigue, pain, and sleep quality were evaluated by intersession assessment, whereas the variables energy, pain, stress, and self-efficacy were evaluated by intrasession assessment.

### 2.6. Statistical Analysis

Data analyses will be conducted by means of the statistical program SPSS v25 and MPlus 7.0. Initially, descriptive statistics will be calculated for all measures of the study (general, clinical characteristics/screening, and primary/secondary/additional secondary outcomes). Subsequently, multivariate analyses will be conducted to define the effect size of the intervention at the different assessment periods: CSA (pre, during, post, and follow-up) and EMA (during). Continuous variables will be analysed using the Kolmogorov–Smirnov test to assess normal distribution and the Levene test for equal variances. First, we will use the Student’s t-test to examine the baseline between-group differences in sociodemographic and clinical characteristics. The primary between-group analysis to assess the effect of treatment will be conducted on an intention-to-treat (ITT) basis with the FIQR total score as a continuous variable and assuming data missing at random. This involves using 2 × 4 mixed-effects ANOVA with the group (TAU+NAT-FM vs. TAU) as the between-groups factor and phase (pre, post, follow-up+6, follow-up+9) as the within-subjects factor. Corrected *ηp*^2^ (partial eta-square) will be estimated for the full model (group and phase main effects and group x phase interaction). We will also report the effect size (Cohen’s *d*) for each pairwise comparison, using the pooled baseline SD to weight the differences in the pre-post means and to correct for the population estimate. Separate models will be estimated for each of the secondary outcomes using the same analytical strategy. 

In addition, to assess the clinical significance of the improvement in the primary outcome (FIQR), we will classify participants into two categories (responders vs. non-responders to treatment), using the following criterion: ≥20% reduction in the pre–post FIQR total scores [60]. This classification will be used to compute the number needed to treat (NNT) in NAT-FM compared with TAU. Finally, we will examine whether the effect of NAT-FM on the primary outcome at post and at the 6 and 9-month follow-ups is mediated/moderated through pre–post changes in secondary measures and personal factor variables (age, years from diagnosis, severity), using a structural equation model. According to the proposal of Luciano et al. [60], change scores before and after NAT-FM treatment will be calculated for these variables, as well as change scores before follow-up for the outcome variables. The analysis of temporality will increase the possibility of drawing conclusions about causality relationships. In this process, the data of the patients of the treatment with the greatest participation in the intervention will be analysed, defining participation as a minimum of 75% attendance of the sessions (per protocol analysis). The Benjamini–Hochberg correction for multiple comparisons will be applied; a procedure to detect false findings designed to overcome the limitations of other common procedures [61].

Complementary analyses will be conducted. Firstly, EMA intrasession measurements obtained from the NAT-FM treatment group will be subjected to a 2 (pre vs. post) × 6 (type of activity) mixed-effects ANOVA for pain, fatigue, self-efficacy, stress, and affect in order to assess the short-term impact of the planned activities. Secondly, EMA between-session measurements obtained from both arms of the study will be analysed both on a time-domain (single regression linear model for each dependent variable) and a frequency-domain basis (spectral analysis for each dependent variable) in order to assess transfer of the treatment effects to daily life.

## 3. Discussion

The main objective of this article is to report the design characteristics of the NAT-FM treatment in comparison with TAU. The protocol for this RCT was designed using SPIRIT recommendations and recorded in a clinical studies database (ClinicalTrials.gov) in accordance with CONSORT guidelines. If the findings of the RCT are sufficiently robust, NAT-FM treatment could be offered as an add-on intervention to the usual treatment (TAU), with a coherent strategic approach that is adjustable to the health resources allocated to FM patients. In this RCT, in addition to evaluating the clinical effects of NAT-FM treatment in the mid- and long-term, we will seek to recognize relevant moderators and mediators of clinical change.

The possible withdrawal of patients from the trial will be one of the risk components that will be considered. In this regard, two strategies will be developed: (a) conduct a sensitivity analysis to determine the impact of adherence to the NAT-FM protocol on the observed effects and (b) combine this RCT to the MOTI-NAT programme (Motivational intervention for Nature Activity Therapy) [62], which will have as its main objective the development of specific therapeutic adherence strategies for patients linked to the different protocols of the NAT Project based on the increased motivation to participate in these therapies. 

NAT-FM treatment is presented as the first intervention that integrates CBT, PNE [27], ET, and nature exposure. The integrative commitment of this RCT is based on the recognition of the scientific evidence identified in different studies in the components of treatment. The results obtained in the proof of concept study [41], designed as a preparation for this protocol, suggest moderate to strong size effects of the therapy, which constitutes an empirical basis for the rationale underlying the intervention.

The results derived from the planned full trial are expected to constitute a preliminary step towards the development of this new generation of therapeutic treatments for intervention in various health problems based on CBT combined with PNE and ET in natural contexts. A deeper understanding of the specific therapeutic effects of NAT-FM could be very informative when considering the relevance of this approach as a complementary model of health intervention, specifically in the domain of chronic pain. With regards to FM, it is worth remembering that there are currently no curative treatments and that developing new approaches that improve the functionality of patients would be of great benefit both at a social and personal level (due to the high impact and prevalence of this condition), particularly given the healthcare (high use of public resources) and economic (high consumption of health resources/job losses) consequences of this illness.

## 4. Conclusions

This study represents the first attempt for combining pain neuroscience education, exercise therapy, psychological therapy, and exposure to nature in the treatment of fibromyalgia.
A randomized controlled two-arm trial is planned to address safety and potential efficacy of NAT-FM (Nature Activity Therapy for Fibromyalgia) in comparison with traditional care (TAU).NAF-FM combines the classical structural assessment (CSA) and ecological momentary assessment (EMA) to obtain more precise information about the dynamics of the variables to be evaluated, to record the affective and cognitive impact of each activity, and to identify its transfer to everyday life.As indicated in previous studies, there is scientific evidence that the improvement in functional status of the people diagnosed with fibromyalgia can be attained by the direct intervention in processes such as the positive and negative affect, self-efficacy, pain, fatigue, emotional regulation, and catastrophising.Some issues arise from the complexity of combining the components of the intervention, particularly those related to logistics and the diversity of activities to be completed by the participants, and the determining the ‘effective’ treatment components.

## Figures and Tables

**Figure 1 ijerph-17-00634-f001:**
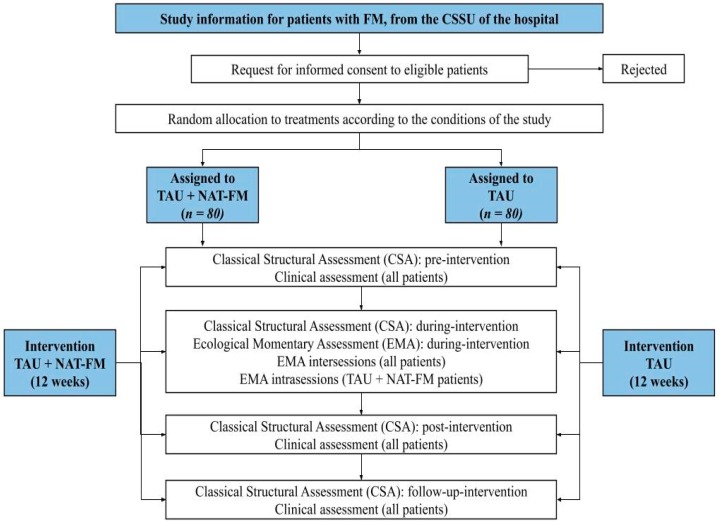
Flow chart of the randomized controlled trial (RCT).

**Table 1 ijerph-17-00634-t001:** Structure of the Nature Activity Therapy for people with Fibromyalgia (NAT-FM), indicating the activities to be performed and the psychological targets to be addressed in each of the sessions.

Session	Activities					Targets					
	NW	YO	HK *	PH	SY **	Negative Affect	Positive Affect	Pain/Fatigue	Emotional Regulation	Self-Efficacy	Catastrophising
1			X					1–16			
2		X	X				1–6	1–2			
3		X	X	X				3–4		1–6	
4	X		X					5–6		7–10	
5	X		X	X				7–8			
6	X	X	X			1–6		9–10			
7	X		X		X			11–12			1–6
8	X		X	X				13–14	1–5		
9		X	X					15–16	6–8		
10			X	X		Family session					
11	X		X		X	1–6			1–8		1–6
12	X		X				1–6			1–10	

Note. The “X” expresses the sessions in which each activity is planned. The numbers of the targets’ columns are referred to the steps described on Table 2. Cognitive behavioural intervention on primary and secondary outcomes is distributed along the sessions. * Hiking is a homework assignment to do as a therapeutic exercise with cognitive targets. ** Shinrin Yoku is understood as mindfulness in a natural context. Nordic walking (NW); yoga (YO); hiking (HK); photography (PH); Shinrin Yoku (SY).

**Table 2 ijerph-17-00634-t002:** Steps in pain neuroscience education and in the cognitive-behavioural treatment for each outcome.

**Pain/Fatigue (Pain Neuroscience Education)**	
1	Disassembling beliefs.
2	Concept of central nervous system and centre sensitization.
3	Concept of pain, fatigue, and pain system.
4	Acute pain vs chronic pain: Purpose of acute pain and how it originates in the nervous system.
5	Pain and damage.
6	Pain neuromatrix theory and virtual body representation.
7	Nociception, nociceptors, action potential, peripheral sensitization, synapses.
8	Descending/ascending pain pathways, spinal cord.
9	Danger signs: modulation and modification.
10	Hyperalgesia and allodynia, hypersensitivity of nervous central system.
11	The role of the brain, pain memory, pain perception, autoimmune evaluation error.
12	Etiology.
13	Relationship with stress and emotions.
14	Relationship with attention, perceptions, pain cognitions, and pain behaviour.
15	Neuroplasticity and how pain becomes chronic.
16	Re-education, graded activity, and exercise therapy
**Self-Efficacy**	
1	Understand the concept of self-efficacy.
2	Recognize the relationship between self-efficacy, pain, and fatigue.
3	Recognize the importance of adapting self-efficacy to real capacity.
4	Become aware of the process of self-efficacy elaboration.
5	Recognize the dynamics of self-efficacy.
6	Identify the sources of self-efficacy.
7	Identify the biases in the creation of self-efficacy.
8	Learn how to stop and replace the biases in the creation of self-efficacy.
9	Relationship between self-efficacy and self-esteem.
10	Transfer: How to work self-efficacy in day life.
**Positive Affect**	
1	Understand the concept of positive emotions.
2	Recognize the relationship between positive emotions, pain, and fatigue.
3	Recognize the importance of actively searching for sources of positive emotions.
4	Identify sources of positive emotions in the context of the sessions.
5	Learn to pay attention to stimuli/conditions that generate positive emotions.
6	Transfer: How to work positive affect in everyday life.
**Negative Affect**	
1	Understand the concept of negative emotions.
2	Recognize the relationship between negative emotions, pain, and fatigue.
3	Recognize the adaptive function of negative emotions.
4	Assess the importance of emotional regulation to reduce negative emotions.
5	Learn to pay attention to the stimuli/conditions that generate negative emotions.
6	Transfer: How to reduce negative affect to everyday life.
**Emotional Regulation**	
1	Understand the concept of emotional regulation.
2	Recognize the relationship between emotional regulation, pain, and fatigue.
3	Identify the 9 types of cognitive regulation of emotions.
4	Identify the relationship between emotional regulation, pain, and fatigue.
5	Learn to pay attention to emotional regulation.
6	Learn to identify the type of emotional regulation usually employed in everyday life.
7	Learn to stop and subtract inappropriate emotional regulation for a proper one.
8	Transfer: How to work emotional regulation in everyday life.
**Catastrophising**	
1	Understand the concept of catastrophism.
2	Recognize the relationship between catastrophism, pain, and fatigue.
3	Learn to recognize the catastrophic thoughts.
4	Learn to pay attention to the catastrophic thoughts.
5	Learn to stop and replace the catastrophic thoughts.
6	Transfer: How to work catastrophising on an everyday basis.

Note. To present in a most understandable way this information to patients, a power point has been used with pictures, examples, and metaphors, according to the recommendations [18]. All these aspects have been reinforced point by point in each session with the book *Explain Pain* in Spanish [27].

**Table 3 ijerph-17-00634-t003:** Measurement scheme and timings for the NAT-FM treatment protocol.

	Pre	During (Session 6)	Post	Follow-Up (6 Months)	Follow-Up (9 Months)
**Classical Structural Assessment (CSA)**					
General information measures					
Sociodemographic information (age, education level, etc.)	X				
Clinical information (medical history, FM history, etc.)	X				
Measures of clinical features and screening					
SCID-I (depression module)	X				
Measures of primary outcome					
FIQR (functional status)	X	X	X	X	X
Measures of secondary outcomes					
PANAS (negative and positive affect)	X	X	X	X	X
CERQ (emotional regulation)	X	X	X	X	X
PPCS (perceived competence)	X	X	X	X	X
RSES (self-esteem)	X	X	X	X	X
PCS (catastrophising)	X	X	X	X	X
PSS-4 (stress)	X	X	X	X	X
Measures of additional secondary outcomes					
HADS (anxiety and depression)	X	X	X	X	X
PIPS (psychological inflexibility)	X	X	X	X	X
TSK-11 (kinesiophobia)	X	X	X	X	X
SF-36 (physical function)	X	X	X	X	X
UKK (functionality)	X	X	X	X	X
AEAC (adverse effects) *			X		
PGIC/PSIC (impression of change) *			X		
**Ecological Momentary Assessment (EMA)**					
Measures of other secondary outcomes					
SAM (emotional state)		X			
VAS (fatigue, pain, and sleep quality)		X			
VAS (self-efficacy and stress)		X			

Note: “X” expresses the moment when each assessment instrument is planned to be administered. Structured Clinical Interview for DSM Axis I Disorders (SCID-I); Revised Fibromyalgia Impact Questionnaire (FIQR); Positive and Negative Affect Schedule (PANAS); Cognitive Emotion Regulation Questionnaire (CERQ); Personal Perceived Competence Scale (PPCS); Rosenberg Self-Esteem Scale (RSES); Pain Catastrophizing Scale (PCS); Perceived Stress Scale (PSS); Hospital Anxiety and Depression Scale (HADS); Psychological Inflexibility in Pain Scale (PIPS); Tampa Scale for Kinesiophobia (TSK-11); 36-Item Short Form Survey (SF-36); UKK Walk Test (UKK); Adverse Effects Assessment Checklist (AEAC); Patient Global Impression of Change (PGIC); Pain Specific Impression of Change (PSIC); Self-Assessment Manikin (SAM); Visual Analog Scale (VAS). * Only for NAT-FM arm.

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
