# Peer review of "Efficacy of a Multicomponent Intervention for Fibromyalgia Based on Pain Neuroscience Education, Exercise Therapy, Psychological Support, and Nature Exposure (NAT-FM): Study Protocol of a Randomized Controlled Trial"

_ijerph, 2020, doi:10.3390/ijerph17020634_

Round 1
Reviewer 1 Report
Thank you for giving me the opportunity to review this interesting manuscript. I consider it timely and novelty in the field of FM. However, I have some minor points that authors should be consider.
Line 147 to 154 and Table 1. Why these activities were selected? Why IO is only practised 4 times or SY twice? Please clarify these aspects.
Line 189 to 192: Please clarify how much session will be performed per week.
Line 343: Why 20% of reduction in the pre-post FIQ-r??
Statistical Analysis: Since multiple comparisons will be made, will p-value be adjusted? (bonferroni, etc.?).
I find a lot of abbreviation in the manuscript which difficulty the read. I encourage authors to consider removing some of them for better clarity.
Reference 41. I think
Author Response
Thank you very much to the reviewer 1 for his/her comments. The requested changes were included in the new version of the paper:
On page 7 (lines 175-180) we explain why we selected the activities listed in Table 1 and we also clarify why we decided to do that number of activities to work on each target. On page 7 (line 196) we explain, in parentheses, that there will be 1 session per week. On page 11 (line 350) we introduce a reference [60], which justifies the criterion for establishing clinical improvement in FIQ-R. On page 12 (line 360-361) we explain that we will apply the Benjamini-Hochberg correction for multiple comparisons, a procedure to detect false discovery designed to overcome limitations of other usual procedures. The authors prefer to keep the abbreviations as presented in the current version of the paper. However, to make it easier to read and follow the abbreviations, we introduced the section "abbreviations" on page 13 (lines 417-421). Fixed on page 15 (line 533-535).Reviewer 2 Report
The study design is well presented and sounds scientifically relevant. I just doubt about the June 2020 deadline; however, as far as I am concerned, I do not have any particular comment and it can be published as its.
Author Response
Thank you very much to the reviewer 2 for his/her comments. We confirm that the deadline reported in the protocol NAT-FM is in accordance with the times established in the project.
Reviewer 3 Report
The manuscript of Serrat et. Al is a study protocol of a randomized controlled trial about a multicomponent intervention for fibromyalgia based on pain neuroscience education (PNE), exercise therapy (ET), psychological support (PS) and nature exposure NAT-FM.
The present manuscript is well written but it is a protocol that will be proportionated to a pilot study. Furthermore the main procedure (NAT-FM) used during the therapy has not been published. Personally, I do not see the need to publish a treatment protocol in a journal with impact factor unless a new methodology would be described or it will proportionated to a large scale study.
In any case, the article about the NAT-FM design and procedures is in preparation, the article is well written and the process and design of NAT-FM is quite well described. Minor Issues:
Introduction:
ET, cognitive behavioral therapy, treatment as usual should be briefly described line 57: substitute 11.000 million with “11 billion” “Patients in both arms will receive TAU, as NAT-FM will, in principle, be complementary treatment to that usually provided by the Spanish National Health System”. Authors should put a referenceTable 1.: change “Ioga” with “Yoga”
Author Response
Thank you very much to the reviewer 3 for his/her comments. All requested adjustments were included in the new version of the document:
On page 2 (line 57) "11.000 million was replaced by “11 billion" and on page 3 (line 102) the reference to the usual treatment for FM in the Spanish National Health System was included. Fixed. On page 4 (line 157), "Ioga" was replaced by "Yoga".Reviewer 4 Report
Review Report
The purpose of the manuscript Efficacy of a Multicomponent Intervention for Fibromyalgia Based on Pain Neuroscience education, Exercise Therapy, Psychological Support, and Nature Exposure (NAT-FM): Study Protocol of a Randomized Controlled Trial, is to describe the design, methods and objectives of the randomized controlled Trial NCT04190771, which will test the efficacy of the Nature Activity Therapy for people with FibroMyalgia (NAT-FM) program compared to treatment as usual. I commend the researchers for taking on this topic as fibromyalgia poses significant treatment challenges. I further commend the authors for using the SPIRIT guidelines to present their study protocol.
Overall, the paper is well written and includes enough detail that the study could be replicated. I have a few comments, based on the SPIRIT checklist that I will address my concerns within each section.
Introduction
The authors present compelling evidence for the need for improved care for fibromyalgia. However, beginning in the 2nd paragraph (line 63) through the 3rd paragraph (line 83), I lose clarity regarding why this particular treatment is being advanced. For example, in the 2nd paragraph, evidence for each of the intervention components (ET, CBT, PNE) are provided. But why do they need to be combined if they are effective on their own? Is it that they are not very effective or they show response variability and they need to be combined for greater benefits? Elaborating on why this new combined treatment is needed and how it is better than current treatments using these same components will help justify to the reader the need for this study and spark interest in your forthcoming results. What is treatment as usual for fibromyalgia and why was this chosen as the comparator?
Methods
Inclusion criteria states the 2010/2011 ACR diagnostic criteria for FM [1,36,37,38], on line 113. Is reference to the 1990 diagnostic criteria here necessary [ref #36]? Some of the information provided under the Intervention group (p.4) would be useful information in the Introduction to give the reader more context about the combined intervention (see comment 1). Concepts presented in Tables 1 and 2 could be summarized and/or presented in text to assist with readability. The formatting in Table 1 could be altered to improve clarity by adding a distinguishing feature (line/border) between where the “Activities” columns end and the “Targets” columns begin. The paragraph beginning on line 181 seems like it belongs under Study Measures. Do these assessments apply only to the Intervention group? If not, then they would be better situated within another section. Will medication use be assessed during the trial? I think line 290 is supposed to read “Ecological” not “Classical”. Line 293 says “were taken”. Does this mean these assessments have already occurred? Again in line 295 “patients had to respond…” sounds like these assessments already took place.Conclusions
Conclusion point #4 could be restated to indicate this conclusion is based on previous literature. I am not sure this is relevant yet to this study, as this study per se has not shown this yet. I agree with Conclusion point #5 and would add that it is also complicates determining the ‘effective’ treatment components. This brings me back to the Introduction and rationale, is it that there will be a greater effect from the combined treatment elements and that is why this study is needed, because alone there is a small effect to individual components, but combined it is hypothesized treatment outcomes will be improved?Author Response
Thank you very much to the reviewer 4 for his/her comments. All requested changes were included in the new version of the manuscript:
In response to the reviewer 4's request, we decided to incorporate an explanatory sentence on the scientific relevance of assessing the synergistic effects of the therapeutic ingredients combination (see below the new text inserted on p. 2: lines 77-80). The usual FM treatment we set as a comparator is described in section 2.4. For this reason, we consider that it is not necessary to mention it again in the introduction, to avoid redundancy (see below the text inserted on page 7: lines 201-202). On page 3 (line 114) we decided to remove "reference 36" as it was not necessary. Although we appreciate reviewer 4's suggestion to integrate the information in Table 1 and 2 into the text, we think that, for reasons of readability, we prefer to report it in the current presentation format. The requested adjustment was made in Table 1 (see below the revised table on page 4: lines 157-158). The verb tense in section 2.5.2 was put in future (see below the text inserted on page: lines 297-302). Conclusions. Changes suggested by the reviewer were included in the conclusions section (see below the text inserted on page 13: lines 410-416).